# Challenges in the Management of Sarcopenia in the Primary Care Setting: A Scoping Review

**DOI:** 10.3390/ijerph20065179

**Published:** 2023-03-15

**Authors:** Thinakaran Kandayah, Nazarudin Safian, Shamsul Azhar Shah, Mohd Rizal Abdul Manaf

**Affiliations:** Department of Community Health, Faculty of Medicine, Universiti Kebangsaan Malaysia, Kuala Lumpur 56000, Malaysia; p114950@siswa.ukm.edu.my (T.K.); drsham@ppukm.ukm.edu.my (S.A.S.); mrizal@ppukm.ukm.edu.my (M.R.A.M.)

**Keywords:** challenges, management, sarcopenia, primary care

## Abstract

Background: Sarcopenia is a disease associated with the loss of muscle mass, strength, and function. It affects the elderly in various ways, such as reduced mobility, compromising their daily activities, and even deteriorating metabolic health. Primary care serves as the first point of contact for patients and plays an important role in health promotion and disease prevention. Hence, this review is conducted to identify the challenges in the management of sarcopenia in the primary care setting. Method: In December 2022, a scoping review was conducted using PubMed, SCOPUS, Web of Science, and a manual search, following the Preferred Reporting Items for Systematic Reviews and Meta-Analyses (PRISMA) criteria. We used articles that have been written in English, and relevant articles were then screened, duplicates were removed, eligibility criteria were applied, and studies that met the criteria were reviewed. The keywords challenges, management, sarcopenia, and primary care were included. Result: The initial search generated 280 publications, and 11 articles were included after inclusion and exclusion criteria for this review. In this review, challenges in the management of sarcopenia in a primary care setting are reviewed based on the screening and diagnosis. Conclusions: With an increasing aging population, it is important to understand the challenges in the management of sarcopenia in a primary care setting. Identification of elderly at risk of sarcopenia, followed by referring the affected elderly for confirmation of the diagnosis, is essential to preventing the adverse health effects. The initiation of treatment that comprises resistance exercise training and nutrition should not be delayed, as they are salient in the management of sarcopenia.

## 1. Introduction

With increasing age, the elderly are prone to myriad of health problems that affect their daily lives. The inevitable aging process causes a gradual decline in physical and mental capacity in the elderly (WHO 2022) [1]. According to McPhee et al., (2018) [2], around 40,000 muscle fibers are lost from the quadriceps muscle per year beyond the age of 30. The aforementioned changes increase the tendency of the elderly to suffer from sarcopenia, which originates from the Greek words sarx, which means muscle, and penia, which means loss. The origin and clinical relevance of sarcopenia were first discussed by Rosenberg (1997) [3]. Subsequently, the aforementioned study drove various types of research on sarcopenia globally, especially in the geriatric field. In 2016, sarcopenia was listed as a disease in the 10th edition of the International Classification of Diseases.

In 2010, the European Working Group on Sarcopenia in Older People (EWGSOP) published a consensus on the definition and diagnosis of sarcopenia (Cruz-Jentoft et al., 2010) [4]. In 2018, the consensus was later revised, and the presence of low muscle strength was used to diagnose probable sarcopenia (Cruz-Jentoft et al., 2018) [5]. The presence of low muscle strength and low muscle mass indicates the diagnosis of sarcopenia, and together with low physical performance, severe sarcopenia is diagnosed.

Apart from the EWGSOP consensus, the Asian Working Group for Sarcopenia (AWGS) published a consensus in 2014 and later updated it in 2019 (Chen et al., (2020) [6]. AWGS used similar criteria to EWGSOP to diagnose sarcopenia but used both hand grip strength and physical performance as the screening test and differed in the cutoff values for the measurements due to the ethnicity differences in body size and lifestyle in the Asian population (Pang et al., 2021; Pipek et al., 2020) [7,8]

The prevalence of sarcopenia worldwide ranges from 10% to 27% (Petermann-Rocha et al., 2021) [9]. In the Asian region, the prevalence of sarcopenia varies from 5.5% to 25.7%, with the incidence of sarcopenia being higher in men (5.1% to 21.0%) compared to women (4.1% to 16.3%) (Chen et al., (2020) [6]. Meanwhile, in Malaysia, the prevalence of sarcopenia is reported to be between 28.5% to 33.6% (Ranee et al., 2022; Sazlina et al., 2020) [10,11].

According to a study conducted by Beaudart et al., (2017) [12], elderly people suffering from sarcopenia have a higher mortality rate, functional decline, rate of falls, and incidence of hospitalization compared to those without sarcopenia.

Primary care plays an important role in health promotion and disease prevention, including sarcopenia, as it serves as the first point of contact for the patients and ensures continuity, comprehensiveness of care, coordination, and is people-centered (WHO 2021) [13]. In view of the various negative impacts attributed to sarcopenia, this review is conducted to identify the challenges in the management of sarcopenia in the primary care setting.

## 2. Material and Methods

Scoping reviews have grown popular in health research because of their use in mapping the extent and character of evidence, particularly in complicated issues, and identifying gaps in the scientific literature. This scoping review is prepared according to the five-stage framework developed by Arksey and O’Malley (2005) [14].

Stage 1: Identifying the research question

After reviewing the core topic of discussion, two research questions were developed to obtain the relevant information.

How is sarcopenia managed in a primary care setting?What are the challenges in the screening of sarcopenia in a primary care setting?

Stage 2: Identifying relevant literature

Inclusion criteria

The inclusion criteria used in this review are based on the broad Population-Concept-Context (PCC) recommendations made by the Joanna Briggs Institute (JBI 2015) [15], as shown in the Table 1 below.

### Search Strategy

The literature search strategy started by creating a list of key search terms, as shown below. We conducted a literature search on PubMed, Web of Science, Scopus, and a manual search for studies published from January 2012 to December 2022. Search terms that were used in this systematic literature review are “challenges” AND “management” OR “treatment” OR “diagnosis” OR “screening” AND “sarcopenia” OR “muscular atrophy” AND “primary care”. The search was then conducted following the PRISMA flow, as shown in the diagram (Figure 1).

Stage 3: Study Selection

After initial database searches, a review of titles and abstracts was conducted to ascertain the qualification of the articles using key search terms and inclusion criteria. All records were imported into a spreadsheet software program (Microsoft Excel 365) to detect and remove all duplications, while irrelevant articles were excluded. Three reviewers determined individual article eligibility based on a review of the title, abstract, and full text. The fourth reviewer was assigned to resolve any disagreements that might arise between the other three reviewers.

A quality appraisal was conducted using the Mixed Method Appraisal Tool (MMAT) as shown in Table 2. The MMAT evaluates the quality of qualitative, quantitative, and mixed-method studies. It focuses on methodological criteria and includes five core quality criteria for each of the following five categories of study designs: quantitative, qualitative, randomized controlled, nonrandomized, and mixed methods (Hong et al., 2018) [16].

The review was conducted according to the Preferred Reporting Items for Systematic Reviews and Meta-Analyses (PRISMA) (Page et al., 2021) [17] checklist, as shown in Figure 1.

Stage 4: Charting the data

The important data were sorted and extracted from the selected documents in a spreadsheet. The relevant information was chosen to answer the research questions of this review. The data that was extracted from the included studies included the information of the author, study type, sample size, and challenges in the management of sarcopenia in the primary care setting.

Stage 5: Collating, summarizing, and reporting the results

The following information was then gathered and documented on a standardized form to document related items with the research information, such as authors, title, year of publication, study design, sample size, and challenges in the management of sarcopenia in a primary care setting.

## 3. Results

The search yielded 61 articles from PubMed, 204 from WOS, 8 from Scopus, and 7 from manual searching, resulting in 280 unique hits. After rigorous selection screening, only 11 articles were included in the full-text assessment, as shown in the PRISMA flow diagram (Figure 1). A descriptive summary of the included studies in this review regarding the study design, sample size, and challenges in the management of sarcopenia in a primary care setting is presented in Table 3.

### 3.1. Challenges in the Screening of Sarcopenia

Out of the total of eleven articles, five highlighted the challenges in the screening of sarcopenia in primary care. According to the study conducted by Piotrowicz et al., (2021) [24] in Poland, which involved 73 participants, the SARC-F questionnaire has a limitation in the screening process due to its low sensitivity value of 35%. It was further elaborated in the article that the SARC-F could be used to rule out sarcopenia instead due to its high specificity value of 85.7%. A study performed by Yang et al., (2018) [25] compared the standard SARC-F questionnaire with a 3-item SARC-F and revealed that the standard SARC-F is a better option, with sensitivity and specificity values of 29.5% and 98.1%, respectively, compared to the 3-item SARC-F, which has sensitivity and specificity values of 13.1% and 97.8%. Both articles findings pointed out the limitations of screening tools in terms of sensitivity. This limitation proves to be a challenge for the primary care physician to identify the elderly at risk of sarcopenia and subsequently refer them to secondary or tertiary care.

A qualitative study by Silva et al., (2020) [28] highlighted the issue of a lack of knowledge among nurses in the primary care setting. It was further mentioned that, due to a lack of knowledge, the screening for sarcopenia is affected because the elderly at risk of developing the disease are not identified. Another separate study by Offord et al., (2019) [22] in the United Kingdom showed that identification of sarcopenia among healthcare professionals is low. Apart from that, the study also showed that there is a lack of diagnosis based on the standard guideline. This shows that a lack of knowledge among the health care professionals complicates not only the screening but also the diagnosis of sarcopenia. Findings from a recent study by Hwang and Park (2022) [27] reveal that the risk factors for sarcopenia are rarely identified by primary care health professionals. Apart from not identifying the elderly at risk, a lack of knowledge on sarcopenia also increases the tendency to miss the diagnosis of sarcopenia. This is very important to take note of, as the aforementioned issues may compromise the outcome as the screening and diagnosis occur in tandem.

### 3.2. Challenges in the Diagnosis of Sarcopenia

In this review, seven articles revealed the challenges in the diagnosis of sarcopenia in primary care. A study by Lera et al., (2018) [20] and Lino et al., (2016) [18] showed that, due to the high cost involved in purchasing dual-energy X-ray absorptiometry (DXA), hand grip assessment is a low-cost and feasible alternative for addressing the issue. A study by Xiang et al., (2022) [26] also highlighted the issue pertaining to the high cost. The study shows that diagnostic tools such as DXA or bioelectrical impedance analysis (BIA) may be unavailable in the primary care setting due to the cost involved.

Besides that, Lera et al., (2020) [21] conducted another study that involved 430 participants who used HTSMayor software for mobile devices and computers. The software estimates the appendicular skeletal mass by using an anthropometric equation or DXA measurements according to the Chilean cut-off point. Results show that HTSMayor software has a sensitivity and specificity of 82.1% and 94.9%, respectively, compared to DXA and could be used to help the primary care physician diagnose sarcopenia instead of depending on conventional DXA, which proves to be expensive. Apart from validity and the lower cost, the HTSMayor software is noted to be feasible in the primary care setting, and a similar study could be conducted in other populations by using the prediction equation and cut-off point for their respective populations. Furthermore, the availability of the aforementioned software also contributed to the development of the Clinical Practice Guide for Sarcopenia in Chile. Nonetheless, it was revealed that the HTSMayor software does have limitations in terms of the lower accuracy of anthropometric measurement and that improvements are needed in the future to increase the accuracy. A similar solution for the incorporation of new technology was shown by another study (Merchant et al., 2020) [19]. According to the study, primary care physicians face challenges such as a shortage of time, multidisciplinary resources, and the skills to perform geriatric assessment. In addressing this issue, the application of a mobile app called the RGA app proved to be feasible, time-saving, and easy to use in the primary care setting. Apart from that, the RGA app also offers operational flexibility, as it can be performed by any healthcare professional.

A study by Cheng et al., (2021) [23] involved 1587 participants and was conducted to adjust and cross-validate skeletal muscle mass measurements between the bioimpedance analysis (BIA) and dual-energy X-ray absorptiometry (DXA). The study reveals that bioimpedance analysis (BIA), based on the predicted value of appendicular skeletal mass, overestimated the skeletal muscle mass measurement compared to the DXA measured appendicular skeletal mass. It was reported the prevalence based on predicted ASM from BIA was (40.8%) compared to (39.4%) on DXA-measured ASM. Despite the overestimation, with the adjustment equation, the BIA is a feasible tool for sarcopenia screening in community and clinical settings. Furthermore, in the primary care setting, it is a huge hurdle for the primary care physician to diagnose the elderly with DXA, as it is not easily available and expensive.

## 4. Discussion

From the perspective of a primary care setting, it is important for health care professionals to be aware and knowledgeable in order to choose the appropriate definition and diagnostic algorithm. There are various definitions and diagnostic algorithms that have resulted in differences in sarcopenia detection globally (Pang et al., 2021) [7]. The challenge faced by a primary care physician is the selection of appropriate screening tools that could be used in the process of screening for sarcopenia. Apart from being well versed in the usage of the screening tools, they should also consider the limitations of the screening tools and could decide to combine the screening tools, such as SARC-F and calf circumference measurement, if necessary. For example, according to Chen et al., (2020) [6] and Cruz-Jentoft et al., (2018) [4], the SARC-F questionnaire is recommended to be used in the case-finding process because it is feasible in primary care settings. However, in a separate study by Dedeyne et al., (2021) [29], the limitations of SARC-F as a screening tool for sarcopenia are highlighted. Apart from that, it was also mentioned in the study that assessments for sarcopenia could be conducted without screening. The aforementioned statement differs from the recommendation made in the revised EWGSOP and AWGS that uses the approach of case finding to identify the elderly at risk for sarcopenia. In addition to the option of combining SARC-F and calf circumference measurement, the study by Tanaka et al., (2018) [30] offers other feasible alternatives, such as the Yubi Wakka finger ring test. The elderly people are at increased risk for sarcopenia if the measured calf just fits or is smaller than their finger ring. Furthermore, a study by Ishii et al., (2014) [31] shows that the Ishii screening tool could be used by measuring the total score of age, grip strength, and calf circumference.

The primary care physician’s ability to diagnose sarcopenia is heavily influenced by the type of diagnostic tools used. The use of diagnostic tools should be feasible in primary care settings, as has been highlighted by Chen et al., (2020) [6]. Challenges arise in terms of portability, cost of purchasing, training involved for the staff, and maintenance involved. Hence, these challenges influence the usage of the appropriate diagnostic tool in the primary care setting. In view of the aforementioned limitations, according to the latest consensus of EWGSOP and AWGS, referral of possible sarcopenia patients to the hospital for the confirmation of the diagnosis is recommended. However, the use of portable BIA could be a solution to increase the diagnostic capability of primary care physicians with a careful consideration of the cost involved.

This review also emphasizes the incorporation of new technology in dealing with the high cost of the standard diagnostic tool. This could be useful as it is less time-consuming and paperless in nature. In addition to the advantage of feasibility, the introduction of mobile applications could ensure a standardized usage of screening tools among primary care professionals, as they play an imperative role in identifying the elderly at risk of sarcopenia. Despite the various advantages offered by this mobile application, other factors should be considered, such as patient data privacy and the usage of this mobile application in locations with limited internet connectivity.

The entire primary care team should have adequate knowledge in the management of sarcopenia, not just the doctors. This is vital to ensure optimum management of sarcopenia. Inadequate knowledge could lead to an undesirable outcome as the staff may wrongly screen the elderly and those who are actually at risk of sarcopenia are not identified. Furthermore, primary care physicians should also be familiar with the risk factors for sarcopenia. They are urged to identify the modifiable risk factors associated with sarcopenia, as this will help delay the progression of sarcopenia. A study conducted by Sazlina et al., (2020) [11] in primary care clinics in Malaysia reiterates the importance of identifying modifiable risk factors for sarcopenia, such as physical activity and body mass index.

Extensive research and development in the field of sarcopenia are vital for clinicians in order to understand the disease and improve the treatment and management of sarcopenia. A study conducted by Witham et al., (2021) [32] revealed the establishment of a sarcopenia registry in the United Kingdom. This registry is valuable data for researchers, as they could use it in the recruitment of participants for studies. However, the establishment of a registry in other countries could be challenging as it involves various factors such as the cost, training of staff, and maintenance of the registry.

## 5. Conclusions and Recommendations

Ideally, a primary care physician should initiate the screening for the elderly at risk of sarcopenia and subsequently proceed to establish the diagnosis by using the appropriate diagnostic tools and initiate the treatment. This will prevent delays in the diagnosis and management. However, this is not possible globally due to various factors, such as the portability of the diagnostic tool and the cost involved in purchasing the diagnostic tool. Hence, a primary care physician should be encouraged to screen the elderly at risk and refer for the confirmation of a sarcopenia diagnosis when it is not possible. Apart from using the standard tools, such as the SARC-F questionnaire, validated software could be utilized in the process of screening and diagnosis.

In conclusion, with the increasing aging population globally, challenges in the management of sarcopenia in primary care should be viewed as an opportunity for enhancement of primary care services, rather than a problem, and subsequently avert the undesirable outcome attributed to sarcopenia.

## Figures and Tables

**Figure 1 ijerph-20-05179-f001:**
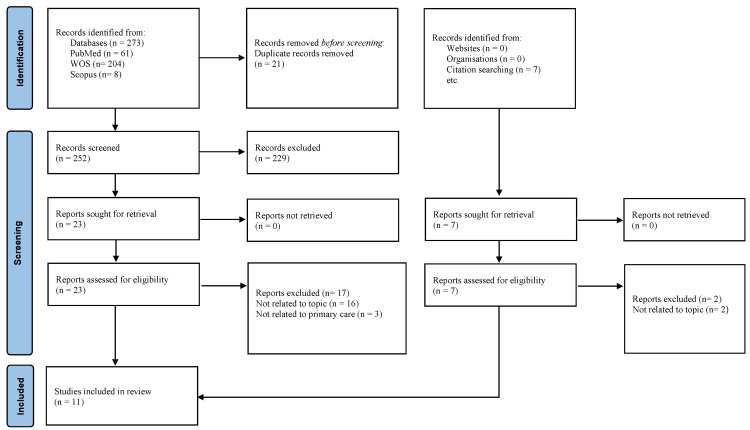
PRISMA.

**Table 1 ijerph-20-05179-t001:** Inclusion criteria.

Population	Concept	Context
(i) Elderly population (ii) Male and female gender	Any challenges in the management of sarcopenia from a primary care perspective from 2012 through 2022.	(i) Research articles are limited to studies written in English language; (ii) Original research articles; (iii) Full text of original articles.

**Table 2 ijerph-20-05179-t002:** The details of the MMAT assessment.

		Is the Sampling Strategy Relevant to Address the Research Question?	Is the Sample Representative of the Target Population	Are the Measurements Appropriate?	Is the Risk of Nonresponse Bias Low?	Is the Statistical Analysis Appropriate to Answer the Research Question?
(Lino et al., 2016) [18]	Quantitative descriptive	Yes	Yes	Yes	Yes	Yes
(Merchant et al., 2020) [19]	Quantitative descriptive	Yes	Yes	Yes	Cannot tell	Yes
(Lera et al., 2018) [20]	Quantitative descriptive	Yes	Yes	Yes	Yes	Yes
(Lera et al., 2020) [21]	Quantitative descriptive	Yes	Yes	Yes	Cannot tell	Yes
(Offord et al., 2019) [22]	Quantitative descriptive	Yes	Yes	Yes	Yes	Yes
(Cheng et al., 2021 [23]	Quantitative descriptive	Yes	Yes	Yes	Yes	Yes
(Piotrowicz et al., 2021) [24]	Quantitative descriptive	Yes	Yes	Yes	Yes	Yes
(Yang et al., 2018) [25]	Quantitative descriptive	Yes	Yes	Yes	Yes	Yes
(Xiang et al., 2022) [26]	Quantitative descriptive	Yes	Yes	Yes	Yes	Yes
(Hwang and Park 2022) [27]	Quantitative descriptive	Yes	Yes	Yes	Yes	Yes
Author	Types of Study	1.1	1.2	1.3	1.4	1.5
		Is the qualitative approach appropriate to answer the research question?	Are the qualitative data collection methods adequate to address the research question?	Are the findings adequately derived from the data?	Is the interpretation of results sufficiently substantiated by data?	Is there coherence between qualitative data sources, collection, analysis, and interpretation?
(Silva et al., 2020) [28]	Qualitative	Yes	Yes	Yes	Yes	Yes

**Table 3 ijerph-20-05179-t003:** Summary of articles.

Author	Article Type	Sample Size	Challenges in the Management of Sarcopenia in Primary Care Setting
(Lino et al., 2016) [18]	Cross sectional	180	Hand grip strength assessment is a feasible and cheaper option in primary care that offers solutions to the usual high cost involved in identifying the risk of sarcopenia.
(Merchant et al., 2020) [19]	Cross sectional	2589	Primary care physicians face the problem of a shortage of time, multidisciplinary resources, or skills to perform geriatric assessment, and the RGA app is a quick and feasible tool that offers a solution to the problem.
(Lera et al., 2018) [20]	Cohort	5250	A dynamometer could be used as a low-cost and feasible tool to identify the elderly at risk for sarcopenia in primary health care and overcome the issue of the expensive and inaccessible method of dual-energy X-ray absorptiometry.
(Silva et al., 2020) [28]	Qualitative	24	Nurses in primary care lack knowledge on sarcopenia.
(Lera et al., 2020) [21]	Cohort	430	(HTSMayor) software serves as an alternative for the expensive and inaccessible DXA in primary care.
(Offord et al., 2019) [22]	Cross sectional	61	Identification of sarcopenia among UK healthcare professionals is low, and there is a lack of diagnosis based on the standard guideline.
(Cheng et al., 2021) [23]	Cross sectional	1587	BIA can be used in a community setting but may overestimate skeletal muscle mass. Prevalence (40.8%) based on predicted ASM from BIA compared to (39.4%) on DXA-measured ASM.
(Piotrowicz et al., 2021) [24]	Cross sectional	73	SARC-F has a limitation in the case finding of sarcopenia due to its low sensitivity (35%). A high specificity (85.7%) of SARC-F could be used to rule out sarcopenia.
(Yang et al., 2018) [25]	Cross sectional	384	The 3-item SARC-F may not be suitable for sarcopenia screening at the community level compared to the standard SARC-F with sensitivity and specificity values of 29.5% and 98.1%.
(Xiang et al., 2022) [26]	Cohort	3829	Diagnostic tools such as DXA or BIA may be unavailable in the primary care setting due to the cost involved.
(Hwang and Park 2022) [27]	Cross sectional	1293	Risk factors for sarcopenia are rarely identified by primary care health professionals. Lack of knowledge about sarcopenia increases the tendency to miss the diagnosis of sarcopenia.

## Data Availability

The data that support the findings of this study are available from the corresponding author upon reasonable request.

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
