# Peer review of "Challenges in the Management of Sarcopenia in the Primary Care Setting: A Scoping Review"

_ijerph, 2023, doi:10.3390/ijerph20065179_

Round 1
Reviewer 1 Report
Kandayah and colleagues have conducted a scoping review whose aims was to identify the challenges in the management of sarcopenia in primary care setting. They used the following keywords: challenges, management, sarcopenia and primary care. Of the 280 publications initially identified, 11 articles were finally included after inclusion and exclusion criteria for this review. They concluded that:
(1) The identification of elderly at risk of sarcopenia and followed by referring the affected elderly for the confirmation of diagnosis is essential to prevent the adverse health effects.
(2) Initiation of treatment that comprises of resistance exercise training and nutrition should not be delayed as they are salient in the management of sarcopenia.
Major comments
Authors point to the importance of early identification of sarcopenia in the elderly, although this is not always possible due to the various factors such as the portability of the diagnostic tool and the cost involved in purchasing the diagnostic tools. There are, however, several resources that can be used in primary care for sarcopenia screening. This review cites only the SARC-F as a diagnostic test, which has good specificity but low sensitivity. However, Nadiaska et al. [1] list nine screening tools for sarcopenia, some as simple to use as the finger-circle test. They particularly point out that evaluation tools combining the SARC-F with other factors (SARC-calf, SARC-F+EBM, etc.), Ishii's screening test, the finger-circle test, etc., can improve diagnosis. Bahat et al [2] point out that the SARC-F is indeed the most recommended tool for screening but has low-moderate sensitivity. They also point out that studies performed recently indicate that its sensitivity can be increased by some attempts and it may be used as a reasonable test to screen frailty as well. With the only SARC-F proposal made by the authors, they do not appear to assist the primary care clinician in screening for sarcopenia.
Although shear-wave ultrasound elastography (SWE) is a technique that is not normally available in primary care centers, perhaps it would be useful to inform clinicians of its good diagnostic ability for sarcopenia. In this regard, Jisook Yi et al. [3] report a sensitivity of 88.9% and specificity of 72.7% in VGG19 architecture. This may be useful for the clinician to refer a suspected sarcopenia patient with an inconclusive diagnosis to primary care.
On the other hand, the authors' proposal noted above was point 2, is too obvious and does not seem to be new to primary care clinicians.
I sincerely believe that a direct search of pubmed can give much more information to primary care clinicians about sarcopenia screening than the current scoping review.
Minor comment
Authors should review the style of the references.
References
[1] Nishikawa H, Asai A, Fukunishi S, Takeuchi T, Goto M, Ogura T, Nakamura S, Kakimoto K, Miyazaki T, Nishiguchi S, Higuchi K. Screening Tools for Sarcopenia. In Vivo. 2021 Nov-Dec;35(6):3001-3009. doi: 10.21873/invivo.12595. PMID: 34697131; PMCID: PMC8627754.
[2] Bahat G, ErdoÄŸan T, Ä°lhan B. SARC-F and other screening tests for sarcopenia. Curr Opin Clin Nutr Metab Care. 2022 Jan 1;25(1):37-42. doi: 10.1097/MCO.0000000000000801. PMID: 34861669.
[3] Yi J, Shin Y, Hahn S, Lee YH. Deep learning based sarcopenia prediction from shear-wave ultrasonographic elastography and gray scale ultrasonography of rectus femoris muscle. Sci Rep. 2022 Mar 4;12(1):3596. doi: 10.1038/s41598-022-07683-6. PMID: 35246589; PMCID: PMC8897437.
Author Response
Reply to the reviewer 1.
Thank you for the precious review and feedback.
We do note about the various screening tools available as what been highlighted by Nadiaska et al. [1] and Bahat et al [2]. However, the aforementioned articles do not elaborate or discuss on the challenges faced by the primary care clinicians which is the main focus and objective of this scoping review. We sincerely believe, the review on alternative screening tools should be discussed under separate review. Our review findings highlighted the challenges when utilizing the SARC-F as it most widely used screening tools as what been recommended by the latest EWGSOP and AWGS consensus. Apart from that, we also highlighted the methods available to improve the screening by combining the SARC-F and calf circumference. We used latest the recommendation by the AWGS and EWGSOP, in order for our review to be generalizable.
We did not include findings on the shear-wave ultrasound elastography (SWE) as a diagnostic tool for sarcopenia as it is not common in the primary care practise and does not fit in the objective of our review which is to focus on the challenges in the management of sarcopenia. We humbly believe, the findings could be used if the direction of the review is seen as a review of literature rather than scoping review.
Besides that, we would like to highlight that, extensive search has been conducted using the search engines specifically on the challenges in the management of sarcopenia. A direct search would only generate general findings on sarcopenia, however limited literature that highlighted on the challenges in the management of sarcopenia specifically in the primary care setting which is the main focus of this review.
Reviewer 2 Report
Well-presented paper coping with a real clinical issue.
The method is perfectly detailed. Congratulations to the authors!
Tables and figures are clearly presented.
However, I was disappointed by 2 concerns:
- Two boxes will be needed to present the mobile apps which were noticed in the text with reference
HTSMayor Software for the Diagnosis of Sarcopenia in Community-Dwelling Older Adults
RGA App orRegence Group Administrators
The opinion of the authors of the scientific basis of the App, their advantages/disadvantages, practicability and cost would perfectly fit with this paper.
- The recommendations are not useful. Do GOs have to proceed in two steps. Screening and then Assessment done with more sophisticated and costly approaches
Author Response
Thank you for the precious review and feedback. Below are the corrections made.
Besides that,(Lera et al. 2020) conducted another study that involved 430 participants and found out that (HTSMayor) software showed good sensitivity and specificity and could be used to help the primary care physician to diagnose sarcopenia instead of depending on conventional DXA that proves to be expensive. Apart from validity and the lower cost, (HTSMayor) software is noted to be feasible in the primary care setting and similar study could be conducted in other population by using the prediction equation and cut-off point for their respective population. Besides that, the availability of the aforementioned software also contributed to the development of the Clinical Practise Guide of Sarcopenia in Chile. Nonetheless, it was revealed that (HTSMayor) software does have limitation in terms of lower accuracy of anthropometric measurement and improvements are needed in the future to increase the accuracy. Similar solution of the incorporation of new technology was shown by another study (Merchant et al. 2020). According to the study, primary care physicians face the challenges such as shortage of time, multidisciplinary resources, and the skills to perform geriatric assessment. In addressing this issue, application of a mobile app that is called as (RGA app) showed to be feasible, time saving and easy to use in the primary care setting. Apart from that, the (RGA app) also offers operational flexibility as it can performed by any healthcare professionals.
Conclusion and recommendation
Ideally a primary care physician should initiate the screening for elderly at risk of sarcopenia and subsequently proceed to establish the diagnosis by using the appropriate diagnostic tools and initiate the treatment. This will ensure prevention of delay in the diagnosis and management. However, this is not possible globally due to the various factors such as the portability of the diagnostic tool and the cost involved in purchasing the diagnostic tools. Hence, a primary care physician should be encouraged to screen the elderly at risk and refer for the confirmation of sarcopenia diagnosis when it is not possible. Apart from using the standard tools such as SARC-F questionnaire, validated software could be utilised in the process of screening and diagnosis.
In conclusion with the increasing aging population globally, challenges in the management of sarcopenia in the primary care should be viewed as an opportunity to enhance the primary care services rather than a problem and subsequently avert the undesirable outcome attributed to sarcopenia.
Round 2
Reviewer 1 Report
Major comments
The authors have basically maintained the same content as in the first version. In my opinion, this study should be improved. It should at least correctly give the information given in the selected articles (see point 8 of minor comments). We understand that an important objective of this review is to inform primary care clinicians about the diagnostic rules for sarcopenia. However, they do not give a single sensitivity or specificity value for the rules considered (see point 7 of minor comments). English should be extensively revised. Some errata are noted in the "minor comments" section.
Minor comment
1. The style should be improved. Thus, for example, in line 9 instead of saying: “Sarcopenia affects ...” should read: “It affects ...”
2. Ln 33: change “muscle fibres” to “muscle fibers”.
3. Ln 35. Phrase: “Greek word, sarx which means muscle …” shouldn't it be “Greek word sarx, which means muscle …”.
4. Ln 57: Change “hospitalisation” to “hospitalization”.
5. Ln 56-58: The conclusion of the Beaudart et al [1] meta-analysis to which the authors refer is not that patients with sarcopenia have higher mortality and physical decline rates than younger patients but that they are higher than those without sarcopenia.
6. Ln 157-158. The authors should better clarify the scope of the HTSMayor software [2]. First, they should indicate that the sensitivities given by the authors have been obtained taking DXA as the gold standard. They should also indicate the estimates of sensitivity (82.1%) and specificity (94.9%). This is an important aspect given the low sensitivity of the SARC-F.
7. Ln 164: Change “Clinical Practise Guide” to “Clinical Practice Guide”.
8. The report given by the authors of the Cheng study [3] is remarkably limited. First, to refer to the diagnostic technique they use only the acronym BIA when it should read "bioimpedance analysis." Then they say that it may overestimate skeletal muscle mass compared to DXA (Table 1 and lines 174-176). Cheng et al however add that (sic) “with adjustment equations, BIA can be used as a quick and reliable tool for screening sarcopenia in community and clinical settings with limited access to better options”.
References
[1] Beaudart C, Zaaria M, Pasleau F, Reginster JY, Bruyère O. Health Outcomes of Sarcopenia: A Systematic Review and Meta-Analysis. PLoS One. 2017 Jan 17;12(1):e0169548. doi: 10.1371/journal.pone.0169548. PMID: 28095426; PMCID: PMC5240970.
[2] Lera L, Angel B, Márquez C, Saguez R, Albala C. Software for the Diagnosis of Sarcopenia in Community-Dwelling Older Adults: Design and Validation Study. JMIR Med Inform. 2020 Apr 13;8(4):e13657. doi: 10.2196/13657. PMID: 32281942; PMCID: PMC7186874.
[3] Cheng KY, Chow SK, Hung VW, Wong CH, Wong RM, Tsang CS, Kwok T, Cheung WH. Diagnosis of sarcopenia by evaluating skeletal muscle mass by adjusted bioimpedance analysis validated with dual-energy X-ray absorptiometry. J Cachexia Sarcopenia Muscle. 2021 Dec;12(6):2163-2173. doi: 10.1002/jcsm.12825. Epub 2021 Oct 4. PMID: 34609065; PMCID: PMC8718029.
Author Response
Dear respected reviewer,
Thank you very much for the valuable comments. Below are the correction made. Apart from the correction we have also done proofreading.
Sensitivity and specificity value added: Ln (126 -134)
According to the study conducted by (Piotrowicz et al. 2021) in Poland that involved 73 participants, the SARC-F questionnaire has a limitation in the screening due to the low sensitivity value of (35%). It was further elaborated in the article that the SARC-F could be used to rule out sarcopenia instead due to its high specificity value of (85.7%). Study performed by (Yang et al. 2018) compared the standard SARC-F questionnaire with a 3 item SARC-F and revealed that the standard SARC-F is a better option with a sensitivity and specificity value of (29.5% and 98.1%) respectively compared to the 3 item SARC-F that has sensitivity and specificity value of (13.1% and 97.8%). Both article findings pointed out the limitation of the screening tools in terms of sensitivity.
Ln 145 Correction on sentence grammar : Findings from a recent study by (Hwang & Park 2022) reveals that the risk factors of sarcopenia are rarely identified by the primary care health professionals.
Ln 148 This is very important to take note as the aforementioned issues may compromise the outcome in sarcopenia detection as the screening and diagnosis occurs in tandem.
Ln 9 : Changed from sarcopenia affects to it affects
Ln 33 : Changed from “muscle fibres” to “muscle fibers”.
Ln 35 : Changed from “Greek word, sarx which to “Greek word sarx
Ln 57 : Changed from “hospitalisation” to “hospitalization”
Ln 56-58: Correction done : According to a study conducted by (Beaudart et al. 2017), elderly people suffering from sarcopenia have a higher mortality rate, functional decline, rate of falls and incidence of hospitalization compared to those without sarcopenia.
Ln 158-162 : Correction done : Besides that,(Lera et al. 2020) conducted another study that involved 430 participants that used (HTSMayor) software for mobile devices and computers. The software estimates the appendicular skeletal mass by using an anthropometric equation or DXA measurements according to Chilean cut-off point. Results shows that (HTSMayor) software has a sensitivity and specificity of (82.1% and 94.9%) compared to DXA and could be used to help the primary care physician to diagnose sarcopenia instead of depending on conventional DXA that proves to be expensive.
Ln 164 : Correction done “Clinical Practise Guide” to “Clinical Practice Guide”
Ln 178 to Line 186 : Correction done : Study by (Cheng et al. 2021) that involved 1587 participants was conducted to adjust and cross-validate skeletal muscle mass measurements between bioimpedance analysis (BIA) and dual-energy X-ray absorptiometry (DXA). The study reveals that bioimpedance analysis(BIA), based on the predicted value appendicular skeletal mass overestimated the skeletal muscle mass measurement (40.8%) compared to the DXA measured appendicular skeletal mass (39.4%). Despite the overestimation, with the adjustment equation BIA is a feasible tool for sarcopenia screening in community and clinical settings. Furthermore, in the primary care setting, it is a huge hurdle for the primary care physician to diagnose the elderly with DXA as it is not easily available and expensive. Table 1 correction done.
Additional corrections
Ln 207 to line 212 : Other options of screening tool besides SARC-F
In addition to the option of combining SARC-F and calf circumference measurement, study by (Tanaka et al. 2018) offers other feasible alternatives such as the Yubi Wakka finger ring test. Elderly is at increased risk for sarcopenia if the measured calf just fits or is smaller than their finger-ring. Besides that, study by (Ishii et al. 2014) shows that Ishii screening tool could be used by measuring the total score of age, grip strength, and calf circumference.
Reference addition :
Ln 288-290 :
Ishii, S., Tanaka, T., Shibasaki, K., Ouchi, Y., Kikutani, T., Higashiguchi, T., Obuchi, S. P., et al. 2014. Development of a simple screening test for sarcopenia in older adults. Geriatrics and Gerontology International 14(SUPPL.1): 93–101. doi:10.1111/ggi.12197
Ln 326-328
Tanaka, T., Takahashi, K., Akishita, M., Tsuji, T. & Iijima, K. 2018. “Yubi-wakka” (finger-ring) test: A practical self-screening method for sarcopenia, and a predictor of disability and mortality among Japanese community-dwelling older adults. Geriatrics and Gerontology International 18(2): 224–232. doi:10.1111/ggi.13163
Proofreading corrections in indicated with ( )
Ln 52 : the prevalence of sarcopenia (is)
Ln 56 : elderly (people)
Ln 57 : According to a study conducted by (Beaudart et al. 2017), elderly people suffering from sarcopenia have a (higher) mortality rate, functional decline, rate of falls and incidence of hospitalization compared to those without sarcopenia.
Ln 65 : Scoping reviews (have grown popular)
Ln 76 : The inclusion criteria used in this review are based (on)
Ln 142 : Apart from that,the study also showed that (there is a lack) of diagnosis
Ln 177 : can (be) performed by any healthcare professionals.
Ln 194 : various (definitions) and diagnostic (algorithms)
Ln 197 : apart from being (well-versed)
Ln 217 : Hence (these) challenges
Ln 231 : The entire primary care team should have adequate knowledge in the management (of) sarcopenia
Ln 242 : (A) study conducted by (Witham et al. 2021) (revealed) the establishment of sarcopenia registry in United Kingdom.
Ln 244 :This registry is a valuable data for researchers as they could use it in the recruitment of participants for studies. (process word deleted)
Ln 256 : utilised to utilized
